# Self-Consistency Preference Optimization

**Archiki Prasad** [1 2]   **Weizhe Yuan** [1 3]   **Richard Yuanzhe Pang** [1]   **Jing Xu** [1]   **Maryam Fazel-Zarandi** [1]   **Mohit Bansal** [2]
**Sainbayar Sukhbaatar** [1]   **Jason Weston** [1 3]   **Jane Yu** [1]

## Abstract

Self-alignment, whereby models learn to improve themselves without human annotation, is a rapidly growing research area. However, existing techniques often fail to improve complex reasoning tasks due to the difficulty of assigning correct rewards. An orthogonal approach that is known to improve correctness is self-consistency, a method applied at inference time based on multiple sampling in order to find the most consistent answer. In this work, we extend the self-consistency concept to help *train models*. We thus introduce self-consistency preference optimization (SCPO), which iteratively trains consistent answers to be preferred over inconsistent ones on unsupervised new problems. We show SCPO leads to large improvements over conventional reward model training on reasoning tasks such as GSM8K and MATH, closing the gap with supervised training with gold answers or preferences, and that combining SCPO with standard supervised learning improves results even further. On ZebraLogic, SCPO finetunes Llama-3 8B to be superior to Llama-3 70B, Gemma-2 27B, and Claude-3 Haiku.

## 1. Introduction

Training large language models (LLMs) on human-annotated data has improved their performance on a wide array of tasks (Bai et al., 2022; Touvron et al., 2023). However, the size and quality of human data remains a major bottleneck as the data collection process is often resource-intensive in terms of cost, time, and expertise. To address this challenge, recent works focus on iteratively training from model-generated data via *self-training* (Yuan et al., 2024; Chen et al., 2024b). Notably, Yuan et al. (2024)

propose a "self-rewarding" training pipeline for instruction-following, comprising two steps: (i) using the LLM to generate new queries and self-evaluating the generated responses for each query; and (ii) building preference pairs and training the LLM using iterative direct preference optimization loss (DPO; Rafailov et al., 2024; Xu et al., 2023). However, Huang et al. (2024) demonstrate that LLMs struggle at evaluating the correctness of their own responses on complex problem-solving tasks which have *an unambiguous correct answer*, thereby rendering Yuan et al.'s self-evaluation approach ineffective. Using an external reward model (RM) to rank responses can have similar problems; even if such models are trained on reasoning tasks they may still suffer on out-of-distribution problems (Casper et al., 2023; Zhang et al., 2024; Mahan et al., 2024).

To address this, we introduce *Self-consistency Preference Optimization* (SCPO). SCPO is an approach to self-train LLMs for complex problem-solving tasks without access to gold solutions or final answers in the training data. Our approach leverages the concept of self-consistency (Wang et al., 2023), an inference-time only approach that improves performance on reasoning tasks by generating multiple solutions using the LLM and choosing the most frequent final answer. More consistent answers are more likely to be correct because mistakes made by the model are often random, so incorrect solutions are unlikely to lead to the same answer multiple times (Fischler & Bolles, 1981; Chen et al., 2023). In SCPO, the self-consistency concept is instead applied *during unsupervised self-training*. The method consists of (i) *selecting* model-generated queries, (ii) *annotating* preference pairs using the most self-consistent response (winner) and least self-consistent response (loser), and (iii) *optimizing* a loss function that is weighted for each instance depending on the model's confidence in the preference pair. Additionally, we propose a *semi-supervised* variant of SCPO that jointly trains LLMs on labeled and unlabeled instances, taking advantage of human annotations whenever available. Unlike self-consistency applied during inference, SCPO does not increase inference-time compute, but they can also be combined together for better performance.

In our experiments using Llama-3 8B models (Dubey et al., 2024), we show that even without access to *any* gold answers during training, two iterations of unsupervised SCPO

---

[1]Meta FAIR [2]UNC Chapel Hill [3]New York University. Correspondence to: Archiki Prasad <archiki@cs.unc.edu>.

*Proceedings of the 42nd International Conference on Machine Learning*, Vancouver, Canada. PMLR 267, 2025. Copyright 2025 by the author(s).

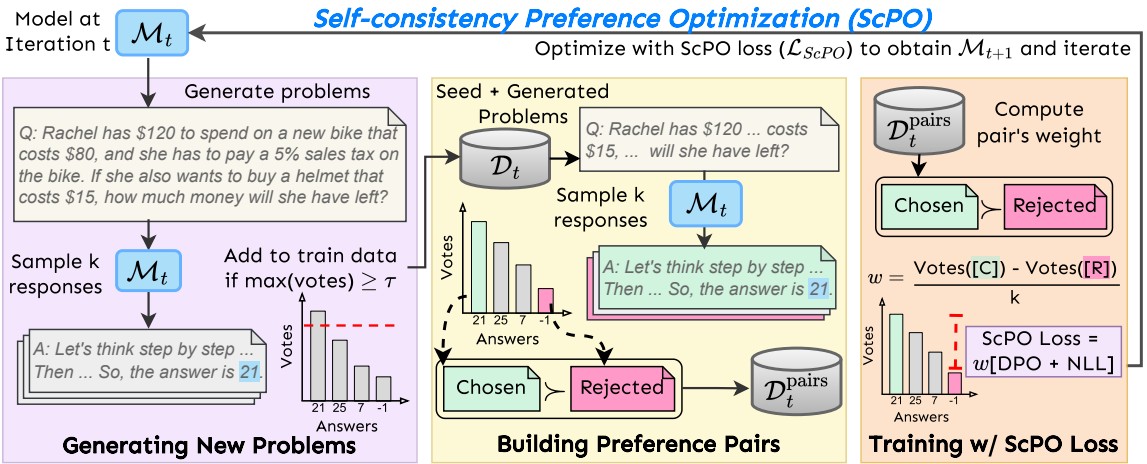

Figure 1. **Self-consistency Preference Optimization** (SCPO). Given a query, we sample multiple responses from the current model $\mathcal{M}_t$ and count the frequency of each answer (i.e., votes). We select the highest and lowest votes as chosen and rejected responses (middle), and use these preference pairs to train the model with weighted $\mathcal{L}_{\text{SCPO}}$ loss (right). We employ a similar pipeline for generating new queries from the model itself (left), filtering out data where self-consistency is low.

improves zero-shot accuracy of the base model by $22.74\%$ and $5.26\%$ (absolute) on GSM8K (Cobbe et al., 2021) and MATH (Hendrycks et al., 2021) respectively, closely matching the performance ($< 1\%$ difference) of the *supervised* baseline from Pang et al. (2024). Moreover, when supplied with the gold labels in the training set and additional model-generated problems, semi-supervised SCPO improves GSM8K accuracy over the supervised baseline by $2.35\%$. On challenging logical puzzles in ZebraLogic (Dziri et al., 2024) – where only test puzzles (without solutions) are publicly available – training Llama-3 8B with SCPO improves puzzle accuracy by $6.5\%$, outperforming larger LLMs such as Llama-3 70B, Gemma-2 27B (Team et al., 2024), and Claude-3 Haiku (Anthropic, 2024).

## 2. Self-consistency Preference Optimization

As depicted in Figure 1, SCPO is an unsupervised iterative training method that starts with a base language model. Each iteration makes use of existing training problems/queries (without labels) as well as newly generated problems. The self-consistency metric is used in both generating new problems and building preference pairs. We describe each step of SCPO's iterative training setup below. All prompts for solution generation and new problem generation can be found in Appendix D.

**Initialization.** SCPO assumes access to an initial base model $M_0$ and a small amount of (seed) high-quality unlabeled queries, which are typically complex reasoning problems. The model will be trained and updated at each training iteration resulting in models $M_1, M_2, \cdots, M_T$, where $T$ is the total number of iterations. Instead of gold labels (an-

swers) for responses, SCPO uses the consistency of the model $M_t$, as measured by a real-valued vote function $\mathcal{V}(\cdot)$ defined below, to rate and rank the quality of each response. Our vote function is based on *self-consistency* (Wang et al., 2023) of the model. In fact, SCPO can also be used with *any* measure of model consistency such as internal consistency (Liang et al., 2024) or universal consistency (Chen et al., 2024a).

**Generating New Problems.** Following other self-alignment methods (Yuan et al., 2024; Yu et al., 2024), we use few-shot prompting to self-generate additional problems from the model. Using the seed set, multiple example problems are chosen at random and placed in context to generate a new problem. Note that some prior works are constrained to simultaneously generating both a new query along with its corresponding correct answer (Yu et al., 2024). In contrast, with SCPO, we do not rely on accurately generating the corresponding answer, allowing the model to generate more diverse problems as long as the problems are *well-formed* and at least some are *answerable*. While the model may generate some unanswerable queries, these can be filtered out using the vote function $\mathcal{V}(\cdot)$. Specifically, we filter out query $x$ if none of the responses generated by $M_t$ have vote $\geq \tau$ (shown in Figure 1; left). At each iteration $t$, we augment the seed queries with the problems generated from $M_t$ to obtain the training problems for the next iteration $\mathcal{D}_{t+1}$.

**Building Self-Consistency Preference Pairs.** For each problem $x$ in the training data $\mathcal{D}_t$, we use temperature-based sampling with the current model $M_t$ to generate $k$ responses $\bar{\boldsymbol{y}}_x = \{y_1, y_2, \cdots, y_k\}$ sampled from $M_t(\cdot|x)$ including any rationales, e.g., chain-of-thought (Wei et al.,

2022), followed by the final answer. Following Wang et al. (2023), the vote function $\mathcal{V}(\cdot)$ extracts the final answer corresponding to each response $y \in \bar{\boldsymbol{y}}_x$ via $\mathrm{ans}(\cdot)$ and returns the relative frequency of the final answer, i.e., $\mathcal{V}(y) = \sum_{m=1}^{k} \mathbb{1}(\mathrm{ans}(y_m) = \mathrm{ans}(y))$. As illustrated in Figure 1 (middle), using the vote function, we create preference pairs $\mathcal{D}_t^{\mathrm{pairs}}$ by selecting the *most consistent* response as the chosen (winning) response and selecting the *least consistent* one as the rejected (losing) response, provided that the vote of the chosen response is greater than a threshold $\tau$.[1] In other words,

$$\mathcal{D}_t^{\mathrm{pairs}} = \{(x, y^+, y^-) \mid x \in \mathcal{D}_t, y^+ = \arg\max_{y \in \bar{\boldsymbol{y}}_x} \mathcal{V}(y),$$
$$y^- = \arg\min_{y \in \bar{\boldsymbol{y}}_x} \mathcal{V}(y), \text{ and } \mathcal{V}(y^+) \geq \tau\}.$$

**SCPO Loss Function.** SCPO operates under the assumption that when multiple responses sampled for problem $x$ map to the same answer, then the predicted answer is likely to be correct, the same assumption as in Wang et al. (2023). Consequently, we use consistency via a vote function $\mathcal{V}(\cdot)$ as a proxy to create preference pairs. However, at the same time, the number of votes attained by a response can also reflect the model's *confidence* in the response (Xiong et al., 2024; Kabra et al., 2024), implying that pairs where the vote margin – the difference in votes attained by the chosen vs. the rejected response – is larger, are of *higher quality* and vice-versa (refer to Appendix A). We model this in SCPO's training by using an instance-level weight $w(x)$ to the loss, i.e., for the preference pair $(x, y^+, y^-) \in \mathcal{D}_t^{\mathrm{pairs}}$, $w(x) = (\mathcal{V}(y^+) - \mathcal{V}(y^-))/k$, where $k$ is the total number of responses generated for each question (total number of votes cast).[2] We thus use the following loss function:

$$\mathcal{L}_{\mathrm{SCPO}}(y^+, y^- | x) =$$
$$\underbrace{-w(x) \log \sigma \left( \beta \log \frac{M_\theta(y^+ | x)}{M_t(y^+ | x)} - \beta \log \frac{M_\theta(y^- | x)}{M_t(y^- | x)} \right)}_{\text{Weighted DPO Loss}}$$
$$\underbrace{- \frac{\alpha w(x)}{|y^+|} \log M_\theta(y^+ | x)}_{\text{Weighted NLL Loss}}.$$

The loss includes a DPO and NLL term similar to the recently introduced supervised IRPO (Pang et al., 2024) loss, but in our case we have an unsupervised objective and use our introduced weighted loss. Here $\sigma(\cdot)$ denotes the sigmoid function, and $\alpha, \beta$ are hyperparameters of the loss function, and $\theta$ represents the LLM parameters being trained in the current iteration. At the $t^{\mathrm{th}}$ iteration, we use the initialized

model $M_t$ as the reference model in the DPO loss (Rafailov et al., 2024). After training on this loss, the trained model is used to initialize the next iteration, i.e., $M_{t+1} \leftarrow M_\theta$.

**Iterative Training.** Starting with an initial seed model $M_0$, we train a series of models $M_1, M_2$, i.e. for $T = 2$ iterations (we justify this choice in Appendix B). Each model $M_{t+1}$ is trained using $\mathcal{L}_{\mathrm{SCPO}}$ on $\mathcal{D}_t^{\mathrm{pairs}}$, the data generated by the $t^{\mathrm{th}}$ model, defined as follows:

- $M_0$: Seed LLM, initialized with a pretrained LLM (need not be instruction-finetuned).

- $M_1$: Initialized with $M_0$ to generate $\mathcal{D}_0^{\mathrm{pairs}}$ from $\mathcal{D}_0$ (+ new problems) and trained using $\mathcal{L}_{\mathrm{SCPO}}$.

- $M_2$: Initialized with $M_1$ to generate $\mathcal{D}_1^{\mathrm{pairs}}$ from $\mathcal{D}_1$ (+ new problems) and trained using $\mathcal{L}_{\mathrm{SCPO}}$.

This approach is similar to the Self-Rewarding LM training loop (Yuan et al., 2024) except for the fact that we use the model's self-consistency to score responses instead of using the same model as a judge to verify its own correctness, which Huang et al. (2024) show is often challenging. In contrast to other iterative bootstrapping techniques for reasoning (Zelikman et al., 2022; Pang et al., 2024), SCPO does not require access to gold labels such as gold responses or final answers, allowing SCPO to scale beyond the problems from an existing training dataset.

**Semi-Supervised Training with SCPO.** Although SCPO does not require access to gold labels, we can easily incorporate datasets with gold labels in conjunction with unlabeled datasets during SCPO training. To this end, we alter the preference pair creation strategy described in that case. When *gold labels are available* for a query $x_{\mathrm{gold}}$, we sample $k$ responses, and create pairs such that the chosen response $y^+$ is *correct* and the rejected response $y^-$ is *incorrect* (discarding queries where such pairs cannot be created). Since we already know these pairs are of high quality, we set the weight of annotated instances $w(x_{\mathrm{gold}}) = 1$. For queries that do not have gold labels, we use our self-consistency criterion for pair creation and compute the weighted loss for those examples as before. A special case is that if all data is labeled, the loss reduces to the IRPO loss.

## 3. Experimental Setup

**Datasets and Metrics.** We evaluate the effectiveness of SCPO on a range of math and logical reasoning datasets:

- **GSM8K** (Cobbe et al., 2021) contains a train/test split of 7.5K/1.3K grade school math word problems. For the purpose of this work, we split the train set into a train/dev split with 6.7K/0.8K problems respectively. We use the dev split for hyperparameter tuning and checkpoint se-

---

[1] By design, several responses can share a final answer (but for example, their chain-of-thought may be different). So, we cluster the responses by final answer and pick a response at random.

[2] This normalization ensures that weights $w(x) \in [0, 1]$.

lection. The overall data split becomes 6.7K/0.8K/1.3K in the train/dev/test set, respectively. We report performance based on exact match accuracy of the final numeric answer on the test set.

- **MATH** (Hendrycks et al., 2021) is a dataset of challenging high-school math competitions that contains a train/test split of 7.5K/5K problems, respectively. Similar to GSM8K, we reserve 10% of samples from the train set to create a held-out dev set for model selection and hyperparameter tuning, resulting in our final train/dev/test splits with 6.7K/0.8K/5K problems, respectively. We report the accuracy of the final answer on the test set.

- **ZebraLogic** (Dziri et al., 2024) is a logical reasoning benchmark. It is a test set of 1K logic grid puzzles (or Einstein's puzzles) designed as a constraint satisfaction problem (Prosser, 1993). Each puzzle is comprised of $n$ houses with $m$ unique features, resulting in an $n \times m$ table. Given a list of clues, solving the puzzle requires deducing the correct (unique) assignment of values in the table, i.e., a unique value for each feature and house. Evaluation metrics for this dataset are: puzzle accuracy (overall, easy, and hard puzzles) as well as cell accuracy.

**Base Models.** For GSM8K and MATH, we use Llama-3 Base 8B (Dubey et al., 2024) as the seed model $M_0$. We note that the instruction-tuned version may have already been fine-tuned on the gold data from these tasks, so new experimental settings cannot be reliably tested in that case. For ZebraLogic, we use Llama-3 Instruct 8B (Dubey et al., 2024) as the seed model.

**Preference Training Data.** We use the Llama-3 Instruct 8B model to generate additional problems (queries). For GSM8K and MATH, we prompt the model to generate a problem similar to 4-shot examples of problems from the train set. Note that the prompt only requires valid human-written problems and *not* their corresponding answers. We filter out problems where $\max_{i \le k} \mathcal{V}(y_i) < 0.5k$ (or, $\tau = 0.5k$) where $k$ is the number of responses sampled or votes cast for each query. That is, where less than half of the votes go towards the majority answer, which we found to be a good threshold based on the dev set accuracy (see Section 5). Since $M_1$ models tend to be more consistent than $M_0$ (cf. Section 5), for $M_2$ training data, we increase the filtering threshold $\tau$ to $0.7k$ and $0.6k$ on GSM8K and MATH, respectively. For ZebraLogic, we prompt the model to rephrase or perturb features of a puzzle from the dataset in a one-shot manner. Then, we use the underlying model $M_t$ to generate $k = 16$ responses for each question and filter out questions where none of the responses accrue $\tau = 2$ or more votes (exactly matching solutions) for $M_1$ and set $\tau = 0.5k$ for training $M_2$.

**Baselines.** We compare models trained with SCPO in unsupervised (denoted as $\text{SCPO}_{\text{Unsup.}}$) and semi-supervised (denoted as $\text{SCPO}_{\text{Semi-Sup.}}$) settings against the following:

- **Seed model (Zero-shot CoT)**. We compare against the seed model ($M_0$) using zero-shot chain-of-thought prompting (Kojima et al., 2022) generated with greedy decoding and report results with or without inference-time self-consistency (SC; Wang et al., 2023).

- **Supervised Training with Gold Answers ($\text{IRPO}_{\text{Gold}}$).** We use a strong supervised preference optimization method for reasoning tasks (Pang et al., 2024), to serve as an upper-bound on performance for unsupervised training as this uses *gold data* from the train set, which we compare to unsupervised and semi-supervised SCPO. For each query $x$, preference pairs are constructed such that chosen responses are correct and rejected responses are incorrect with $w(x) = 1$.

- **Unsupervised Training with External RM ($\text{IRPO}_{\text{RM}}$).** We propose a new variant of IRPO that we also expect to be a strong baseline. Given the plethora of publicly-available reward models (RMs; Lambert et al., 2024), in the absence of gold labels, off-the-shelf RMs can be used to score a set of responses $\bar{y} \sim M_t(\cdot|x)$ and create preference pairs such that chosen and rejected responses have the maximum and minimum reward, respectively, i.e., $y^+ = \arg\max_{y \in \bar{y}} \text{RM}(y|x)$ and $y^- = \arg\min_{y \in \bar{y}} \text{RM}(y|x)$ with $w(x) = 1$. We use the strongly performing ArmoRM-Llama3-8B model (Wang et al., 2024a) as a reward model.[3]

- **Language Models Self-Improved (LMSI).** Following Huang et al. (2023), we implement LMSI, another unsupervised baseline that uses LLM self-consistency to generate target CoT solutions for problems and iteratively trains the LLM via supervised finetuning, i.e., the NLL loss, differing from SCPO's weighted preference-based loss. Similar to SCPO, we generate additional reasoning problems using the LLM followed by consistency-based filtering (detailed in Section 2).

**Hyperparameters.** When generating multiple response or new problems from the LLM, we sample with temperature of 0.7 and top-$p = 0.9$. For GSM8K and MATH, we set $k = 8$. With every iteration of training, the models become more consistent due to the training objective (see Section 5), thereby, making picking the rejected response harder, i.e., none of the responses are incorrect or all the responses share the same final answer. Therefore, to sample rejected responses, we further generate 8 responses sam-

---

[3] Wang et al. (2024a) use training splits of GSM8K and MATH to train ArmoRM, rendering these datasets highly in-distribution for the RM while ZebraLogic is out-of-distribution (further discussed in Section 5).

*Table 1.* **GSM8K zero-shot accuracy** after training Llama-3 Base 8B with SCPO and baselines, using greedy or 8-way self-consistency (SC)-based inference. The best performance is in *bold*, and second-best is *underlined*. We list train set sizes for each method: "Seed" corresponds to seed problems in the train set, whereas "Gen." indicates additional problems generated by the model (without answers). IRPO$_{\text{Gold}}$, and SCPO$_{\text{Semi-Sup.}}$, highlighted in green , use the gold answers to create preference pairs (when available, indicated with $^{\dagger}$).

| Method | Iter. | Train Data (K) | Test Acc. (%) | |
|---|---|---|---|---|
| | | # Seed / Gen. | Greedy | SC |
| *without access to gold labels* | | / | | |
| Seed model (zero-shot) | $M_0$ | - / - | 41.17 | 51.80 |
| IRPO$_{\text{RM}}$ | $M_1$ | 5.5 / - | 48.67 | 69.98 |
| | $M_2$ | 4.4 / - | 50.11 | 61.25 |
| LMSI | $M_1$ | 5.3 / - | 53.53 | 63.91 |
| | $M_2$ | 1.1 / 5.2 | 56.71 | 62.55 |
| SCPO$_{\text{Unsup.}}$ | $M_1$ | 5.3 / - | 61.03 | **71.49** |
| | $M_2$ | 1.4 / 5.1 | **63.91** | 71.11 |
| *with access to gold labels* | | / | | |
| IRPO$_{\text{Gold}}$ | $M_1$ | 4.4$^{\dagger}$ / - | 61.41 | 72.93 |
| | $M_2$ | 5.7$^{\dagger}$ / - | 64.29 | 72.56 |
| SCPO$_{\text{Semi-Sup.}}$ | $M_1$ | 4.4$^{\dagger}$ / 1.9 | 63.61 | 74.30 |
| | $M_2$ | 5.7$^{\dagger}$ / 4.5 | **66.64** | **74.75** |

*Table 2.* **MATH zero-shot accuracy** after training Llama-3 Base 8B with SCPO and baselines, using greedy or 8-way self-consistency (SC)-based inference. "Seed" corresponds to seed queries in the train set, "Gen." are additional model-generated problems (without answers). IRPO$_{\text{Gold}}$ and SCPO$_{\text{Semi-Sup.}}$, highlighted in green , use gold answers to train (indicated with $^{\dagger}$).

| Method | Iter. | Train Data (K) | Test Acc. (%) | |
|---|---|---|---|---|
| | | # Seed / Gen. | Greedy | SC |
| *without access to gold labels* | | / | | |
| Seed model (zero-shot) | $M_0$ | - / - | 14.46 | 18.20 |
| IRPO$_{\text{RM}}$ | $M_1$ | 6.4 / - | 18.06 | 24.20 |
| | $M_2$ | 6.5 / - | 18.08 | 22.64 |
| LMSI | $M_1$ | 0.6 / 1.2 | 16.78 | 22.92 |
| | $M_2$ | 1.1 / 2.0 | 16.96 | 20.20 |
| SCPO$_{\text{Unsup.}}$ | $M_1$ | 0.6 / 1.2 | 17.36 | **25.70** |
| | $M_2$ | 1.2 / 2.5 | **19.72** | 24.58 |
| *with access to gold labels* | | / | | |
| IRPO$_{\text{Gold}}$ | $M_1$ | 2.7$^{\dagger}$ / - | 18.64 | 26.88 |
| | $M_2$ | 3.0$^{\dagger}$ / - | 20.32 | 26.88 |
| SCPO$_{\text{Semi-Sup.}}$ | $M_1$ | 2.7$^{\dagger}$ / 1.2 | 19.88 | **27.35** |
| | $M_2$ | 3.0$^{\dagger}$ / 2.2 | **20.48** | 26.92 |

pled with a higher temperature of 1.2 to encourage more diverse answers. On ZebraLogic, due to the complex nature of the response (an $n \times m$ table), we find that sampling a response that gets multiple votes is relatively infrequent, so we set $k = 16$ for this task. All models are trained for 10 epochs with a learning rate of 5e-6 (cosine scheduling), and effective batch size of 16. Lastly, we set DPO loss term hyperparameter $\beta = 0.5$ and NLL regularization coefficient $\alpha = 1$. When a dev set is available (e.g., GSM8K and MATH), we use accuracy on the dev set for checkpoint selection (at every epoch). For ZebraLogic, which is similarly challenging to MATH and does not have a train or dev set, for each iteration, we train for the same number of epochs that performed best during MATH training.

## 4. Main Results

### 4.1. Math Reasoning

**SCPO outperforms unsupervised baselines.** Comparing methods on GSM8K, in Table 1, we observe that training with only one iteration of SCPO outperforms the zero-shot seed model and IRPO$_{\text{RM}}$, by 22.74% and 12.36%, respectively, using greedy decoding. Similarly, on MATH (cf. Table 2), two iterations of SCPO$_{\text{Unsup.}}$ yields an improvement of 5.26% and 1.64% respectively compared to the same two baselines. We further note that while IRPO$_{\text{RM}}$ is not given direct access to the gold labels, it uses the ArmoRM, which has been trained on human-annotated step-level data

based on MATH's train set (Lightman et al., 2024; Wang et al., 2024a). Hence, SCPO's improvement over IRPO$_{\text{RM}}$ would likely be larger if the RM had not used in-domain gold labels during training. Overall, we find SCPO has the ability to outperform RMs, especially in out-of-distribution settings. Lastly, in comparison to LMSI, another iterative and unsupervised baseline, two iterations of SCPO$_{\text{Unsup.}}$ outperform that of LMSI by 7.20% and 2.76% on GSM8K and MATH, respectively, when using greedy decoding. This highlights the importance of a weighted preference objective in training LLMs effectively using self-consistency.

**Iterations of SCPO improve reasoning.** From Tables 1 and 2, we observe that two iterations of SCPO consistently improves the LLM's performance when using greedy decoding in both unsupervised and semi-supervised settings compared to one iteration. On GSM8K, greedy test accuracy improves by 2.88%, and 3.03% when using SCPO for unsupervised and semi-supervised training, respectively. Similarly, on MATH, in Table 2, we find that $M_2$ models with SCPO outperforms their $M_1$ counterparts by up to 2.36% in greedy accuracy. This can be explained by models becoming more accurate and consistent after one round of SCPO training (shown in Section 5). Consequently, this allows us to bootstrap from additional problems in the original and generated training data, for which the $M_0$ model did not have a consistent response. However, we find that the accuracy computed using 8-way self-consistency (SC) saturates after the first iteration, sometimes even resulting in a slight decrease compared to $M_1$. This may happen because now that the model is trained to be more consistent there is

*Table 3.* **ZebraLogic test performance** after unsupervised training of Llama-3 Instruct 8B with SCPO, compared to baselines. "Seed" corresponds to original puzzles in the test set, whereas "Gen." indicates additional puzzles generated. *Taken from the `Leaderboard`.

| Method | Train Data (K) | Puzzle Acc. (%) | | | Cell Acc. |
|---|---|---|---|---|---|
| | # Seed / Gen. | Overall | Easy | Hard | (%) |
| Llama-3 Instruct 70B | - / - | 17.2 | 52.1 | **3.6** | 42.9 |
| Gemma-2 27B IT* | - / - | 16.3 | 50.7 | 2.9 | 41.2 |
| Claude-3 Haiku* | - / - | 14.3 | 47.9 | 1.2 | 37.9 |
| $M_0$ (Llama-3 Instruct 8B) | - / - | 11.6 | 40.0 | 0.4 | 39.1 |
| $M_1$ w/ IRPO$_{RM}$ | 1.0 / - | 11.3 | 37.9 | 1.0 | 42.1 |
| $M_1$ w/ LMSI | 0.4 / 2.0 | 16.2 | 51.1 | 2.6 | 45.8 |
| $M_2$ w/ LMSI | 0.4 / 2.0 | 16.8 | 53.6 | 2.5 | 46.9 |
| $M_1$ w/ SCPO$_{Unsup.}$ | 0.4 / 2.0 | 17.0 | 54.3 | 2.5 | **47.6** |
| $M_2$ w/ SCPO$_{Unsup.}$ | 0.5 / 2.2 | **18.1** | **58.2** | 2.5 | 45.2 |

less benefit from applying self-consistency at inference time (see analysis in Section 5). We find that a third iteration of training also shows minimal gains, however if we utilize the (unlabeled) problems from the test set to build preference pairs, we find that we can obtain additional performance boosts on top of $M_2$, as discussed in Appendix B.

**Unsupervised SCPO is comparable to IRPO training with gold labels.** We can compare the unsupervised training of SCPO with the supervised training using gold labels of IRPO in Tables 1 and 2. The results show that SCPO$_{Unsup.}$ *without using any gold labels* can yield comparable accuracy to IRPO$_{Gold}$ on GSM8K and MATH with $< 1\%$ gap in greedy performance and $< 2\%$ gap in accuracy using 8-way self-consistency after two iterations of training ($M_2$). This comparable performance of SCPO$_{Unsup.}$ is likely due to high correlation (0.8 across the datasets) between the vote shares and accuracy on the test set, as further discussed in Appendix A. Note that on tasks that are challenging for the seed model $M_0$, such as MATH, we can only bootstrap a small set of examples from the original set of training problem as compared to IRPO (i.e., only around a quarter of examples obtain a clear majority answer). However, we can offset this gap in training data by generating new problems using few-shot prompting (cf. Section 2) and creating preference pairs using our self-consistency method. This yields improvements during the second iteration.

**Semi-supervised training with SCPO outperforms IRPO.** Lastly, in Tables 1 and 2, we evaluate the semi-supervised version of SCPO *combined with using gold labels*. We find that on GSM8K, SCPO$_{Semi-Sup.}$ improves the greedy accuracy by 2.35% and SC accuracy by 2.19% in comparison to IRPO$_{Gold}$. Similar trends hold on the MATH dataset, where one iteration of SCPO$_{Semi-Sup.}$ outperforms IRPO$_{Gold}$ by 1.24% using greedy decoding. These results show the utility of using SCPO to bootstrap from model-generated problems even with access to a labeled training set.

In Appendix C, we repeat the math reasoning experiments with Llama-3.1 Base 8B and find that while the absolute performance increases, the relative trends among the baselines remain the same – with two iterations of SCPO$_{Semi-Sup.}$ improving the greedy test accuracy of the seed model by 25.32% and 8.66% on GSM8K and MATH, respectively.

### 4.2. ZebraLogic: A Challenging Logical Reasoning Task

**SCPO outperforms unsupervised baselines.** Table 3 reports performance on ZebraLogic of SCPO and various baselines, using greedy decoding. We observe large improvements over the seed model, Llama-3 Instruct 8B ($M_0$) with one iteration of unsupervised SCPO ($M_1$), improving performance by 5.4% and 8.5% in overall puzzle accuracy (exact match of tables) and cell accuracy (match of each cell in the table), respectively. In contrast, unsupervised training of IRPO$_{RM}$ yields only mild gains over the seed model by 3% in cell accuracy and even a slight drop in puzzle accuracy (11.6% to 11.3%). This can be attributed to ZebraLogic puzzles being out-of-distribution for the ArmoRM (cf. Section 5), thus trailing behind one iteration of SCPO by 5.7% in puzzle accuracy and 5.5% in cell accuracy. Moreover, two iterations of SCPO outperform that of LMSI by 4.6% on easy puzzles and 1.3% on overall accuracy. Taken together, training with SCPO for two iterations improves the performance of the seed model by 8 positions on the leaderboard (from 38[th] to 30[th]) with a 6.5% boost in puzzle accuracy and, to the best of our knowledge, is the best 8B-scale LLM on ZebraLogic.

**8B LLM trained with SCPO outperforms larger models.** Comparison of SCPO-trained models to other models in Table 3 demonstrates that SCPO-training after two iterations ($M_2$) outperforms significantly larger models such as Llama-3 Instruct 70B, Gemma-2 27B, and Claude-3 Haiku by 0.9%, 1.8%, and 3.8% in overall puzzle accuracy, respectively. Additionally, we find that models trained using SCPO also yield the highest cell accuracy. We attribute these gains over

*Table 4.* Ablation comparing unweighted loss ($w(x) = 1$) to the proposed weighted loss used in SCPO. SCPO outperforms the unweighted loss in all cases.

| | Method | Train (K) | | Test Acc. (%) | |
|---|---|---|---|---|---|
| | | # Seed / Gen. | | Greedy | SC (8-way) |
| GSM8K | $M_1$ w/ $w(x){=}1$ | 5.3 / - | | 58.53 | 69.07 |
| | $M_2$ w/ $w(x){=}1$ | 1.4 / 5.1 | | 62.62 | 69.90 |
| | $M_1$ w/ SCPO$_{\text{Unsup.}}$ | 5.3 / - | | 61.03 | 71.49 |
| | $M_2$ w/ SCPO$_{\text{Unsup.}}$ | 1.4 / 5.1 | | 63.91 | 71.11 |
| MATH | $M_1$ w/ $w(x){=}1$ | 0.6 / 1.2 | | 15.92 | 25.34 |
| | $M_2$ w/ $w(x){=}1$ | 1.2 / 2.5 | | 18.74 | 25.58 |
| | $M_1$ w/ SCPO$_{\text{Unsup.}}$ | 0.6 / 1.2 | | 17.36 | 25.70 |
| | $M_2$ w/ SCPO$_{\text{Unsup.}}$ | 1.2 / 2.5 | | 19.72 | 24.58 |

larger models to the substantial improvement in solving easy puzzles with SCPO (up to $10.3\%$).

## 5. Ablations and Analysis

**Importance of weighted SCPO loss.** While the results in Section 4 are obtained using the weighted $\mathcal{L}_{\text{SCPO}}$ loss that is a function of consistency, here we compare SCPO using an unweighted loss. More specifically, we train using the same preference dataset created based on self-consistency of responses, but with $w(x){=}1$ in the $\mathcal{L}_{\text{SCPO}}$ loss. In Table 4, we observe that across *datasets* and *iterations*, the weighted loss consistently outperforms the unweighted version. The improvement in accuracy is even more pronounced for the first iteration of training $M_1$, yielding an improvement of $2.5\%$ in accuracy on GSM8K and $1.44\%$ on MATH with greedy inference. Even in the second iteration, $M_2$ models trained with SCPO outperform their unweighted counterparts by roughly $1\%$ on both GSM8K and MATH. This indicates that it is better to take the amount of votes into account when optimizing for consistency, as this indicates confidence in the chosen and rejected labeling.

**Models become more consistent across iterations.** In Figure 2, we analyze how the degree of model consistency varies across iterations. To this end, we measure the vote share $\mathcal{V}(y^+)/k$ of the most consistent response, i.e., chosen response in self-consistency of models trained using unsupervised SCPO. From Figure 2, we conclude that SCPO training increases the consistency of models with each training iteration across different tasks. We suspect this finding stems from three contributing factors: (i) with increasing iterations models become more accurate (Section 4); (ii) additional rounds of preference-optimization decreases model diversity (Kirk et al., 2024); and (iii) training with SCPO effectively distills the SC distribution into the model's single-sample distribution. Additionally, we find that models are more consistent on tasks with higher test accuracy, i.e., on

*Table 5.* Impact of using different thresholds on majority vote to filter training data on MATH. Margin (%) denotes the difference in accuracy of the chosen and rejected response.

| Setting | Margin | # Train | Test Acc. |
|---|---|---|---|
| $M_0$ | - | - | 14.46 |
| $M_1$ ($\tau = 0.1k$) | 18% | 6.7K | 15.44 |
| $M_1$ ($\tau = 0.3k$) | 44% | 2.4K | 16.34 |
| $M_1$ ($\tau = 0.5k$) | 57% | 1.8K | 17.36 |
| $M_1$ ($\tau = 0.7k$) | 68% | 0.7K | 14.76 |

GSM8K the LLM is most consistent and accurate whereas on ZebraLogic it is the least consistent and accurate.

**Impact of consistency-based filtering on constructing preferences.** In Section 3, when generating self-consistency preference data for GSM8K and MATH, we filter out instances where fewer than half of the votes go towards the majority answer, i.e., $\tau = 0.5k$. The choice of this threshold presents a trade-off between the *number of preference pairs* available for training and the *quality of the training data*, and affects the difference (margin) in accuracy of the chosen and the rejected response. Assuming access to the gold answers to measure quality of preference data, in Table 5, we analyze this trade-off on MATH. As the vote threshold increases from $\tau = 0.1k$ to $\tau = 0.7k$, the quality of training preference pairs increases, with the accuracy margin increasing from $18\%$ to $68\%$. On the other hand, the size of the training data decreases from 6.7K pairs to fewer that 700 pairs. Interestingly, Table 5 shows that as we vary the threshold, the performance of the trained model increases till $\tau = 0.5k$ and then decreases. In other words, from $\tau{=}0.1k$ to $\tau{=}0.5k$ the quality of the preference data (or the accuracy margin) takes precedence over the quantity, improving downstream performance by $1.92\%$. However, when we set $\tau{=}0.7k$, we end up with fewer than 700 pairs to train which we suspect is insufficient (in terms of both data size and diversity) to train a model with 8B parameters.

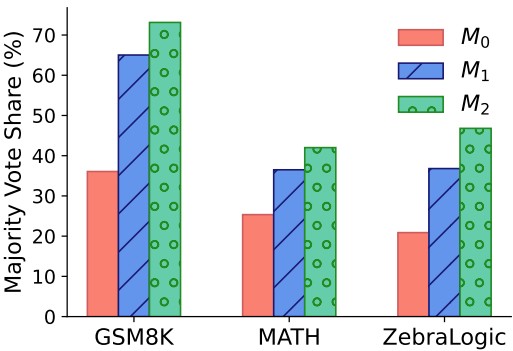

*Figure 2.* Vote share (%) of the most consistent response: $\mathcal{V}(y^+)/k$ increases with iterations across all datasets.

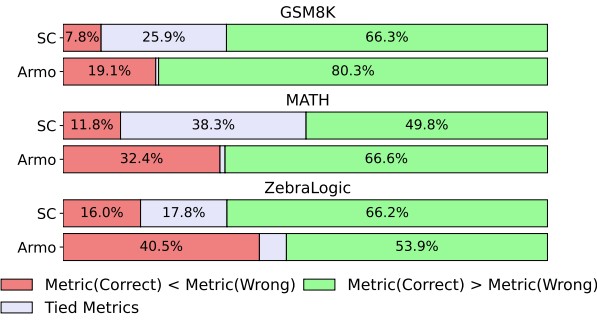

*Figure 3.* Comparing the quality of metrics: self-consistency (SC) and ArmoRM to distinguish between correct and incorrect responses on all datasets.

**Comparison of self-consistency to RMs.** Our results in Section 4 show that models trained with unsupervised SCPO outperform models trained with IRPO using ArmoRM to build preference pairs. To study this further, we conduct additional analysis by measuring the ability of the two methods to distinguish between correct and incorrect responses, comparing the methods to gold labels in Figure 3. We find that ArmoRM consistently has more incorrect orderings of pairwise preferences (the chosen is incorrect and the rejected is correct) than SCPO across all three datasets (shown in red). This added noise in training may be a major factor as to why $IRPO_{RM}$ performs poorly compared to $SCPO_{Unsup}$. On the other hand, self-consistency results in a greater number of ties, i.e., when the chosen and rejected answers get the same number of votes; these are ignored in SCPO's loss since $w(x) = 0$. Lastly, we find in the out-of-distribution setting of ZebraLogic, self-consistency outperforms ArmoRM with 12.3% more correct orderings of pairwise preferences (shown in green in Figure 3).

## 6. Related Work

**Iterative Training of LLMs.** Iterative training or self-training has shown meaningful improvements in a number of domains such as safety (Bai et al., 2022), multilingual reasoning (She et al., 2024), and evaluation (Wang et al., 2024b). Because LLMs often struggle with both generating and validating solutions to complex reasoning tasks, prior works on training LLMs for complex problem-solving tasks largely rely on human-annotated (gold) final answers (Zelikman et al., 2022; Chen et al., 2024b; Pang et al., 2024) or access to an external reward model that performs well on the underlying task (Singh et al., 2024; Dong et al., 2023). However, both these classes of approaches suffer from their own shortcomings. Firstly, manually annotating or verifying the final answer requires working through the solution step-by-step, making it especially resource-intensive for complex multi-step problems. Training strong reward models for such reasoning and problem-solving tasks also often re-

quires human judgements of LLM generations (Cobbe et al., 2021; Uesato et al., 2022; Lightman et al., 2024), making it similarly expensive. Our work focuses on the setting *without access to gold solutions or final answers*, which remains largely unaddressed. While other works such as She et al. (2024); Yuan et al. (2024); Rosset et al. (2024); Tran et al. (2023) geared towards general instruction following tasks (as opposed to reasoning tasks specifically) circumvent the need for human-annotated labels in the dataset by using the model itself to score the responses, these works demonstrate only modest gains in the context of reasoning tasks.

**Consistency in LLMs.** Self-consistency (Wang et al., 2023) relies upon the intuition that sampling several responses, some of which lead to the same answer, lends higher certainty that the consistent answer is the correct one. Application of self-consistency *at inference time* has enabled performance improvements in a number of domains like math (Wang et al., 2023), code generation (Shi et al., 2022; Li et al., 2022; Chen et al., 2018), and even open-ended tasks like summarization and question answering (Chen et al., 2024a). In this work, we explore using self-consistency *at training time* for reasoning tasks, constructing preference pairs according to the self-consistent final answer. While Huang et al. (2023) also use self-consistency to finetune models without access to gold labels via NLL loss, we employ a preference optimization loss function that is weighted according to the consistency of an answer. Intuitively, the consistency of an answer is a reflection of the model confidence, and several prior works have demonstrated that leveraging model uncertainty can lead to faster convergence and improved performance (Gal & Ghahramani, 2016; Krishnan & Tickoo, 2020; Corbière et al., 2019). Concurrently with this work, Jiao et al. (2025) propose training models on "pseudo-feedback" from test cases, wherein they employ self-consistency to construct the test cases itself. However, we note that our work additionally shows the utility of self-consistency in generating new problems to augment the seed data (Section 4) as well as in our weighted loss function (Table 4 in Section 5).

## 7. Conclusion

In this paper, we introduced Self-Consistency Preference Optimization (SCPO). SCPO leverages the concept of self-consistency, usually employed only at inference time, to improve the self-training of large language models. By iteratively optimizing to prefer consistent answers to inconsistent ones, SCPO achieves significant improvements over traditional reward model training without the need for additional gold labels. Our experiments demonstrate the efficacy of SCPO on various reasoning tasks, including GSM8K, MATH, and ZebraLogic, where in the latter it outperforms several larger state-of-the-art language models.

We also showed that SCPO works well in semi-supervised setups with access to some gold labels, in addition to unlabeled inputs – improving performance further. These results highlight the potential of SCPO to improve self-alignment across reasoning tasks – a domain that prior self-alignment methods still struggle with. Future work could extend SCPO to tasks where a single final answer cannot be easily parsed (e.g., summarization) through universal self-consistency (Chen et al., 2024a). While we explore consistency according to several models (Llama-3 and 3.1 8B, Base and Instruct), future work could also investigate consistency according to a suite of other models and tasks.

## Acknowledgements

We sincerely thank Ilia Kulikov, other members of the RAM team at FAIR, as well as the anonymous reviewers for their valuable feedback on the paper. Part of this work was done during an internship at Meta FAIR and was partially supported at UNC by NSF-CAREER Award 1846185, NSF-AI Engage Institute DRL-2112635, DARPA Machine Commonsense (MCS) Grant N66001-19-2-4031. The views contained in this article are those of the authors and not of the funding agencies.

## Impact Statement

This work presents a new training algorithm that uses self-consistency for training large language models on math and logical reasoning tasks without the need for gold labels. The outputs produced by models trained with SCPO may exhibit undesirable behavior similar to the base model and have the same potential for misuse as other fine-tuned LLMs (Weidinger et al., 2021). Hence, more studies are needed to evaluate and mitigate such biases in LLMs.

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

## A. Relationship between Consistency and Accuracy

**Level of consistency or vote share correlates with accuracy.** We observe that the degree of consistency, or vote

share, is positively and strongly correlated with accuracy. This relationship is evidenced in Table 6 by a high rank order correlation for all three datasets, as determined by Somer's D (Somers, 1962), which measures the degree of association between two possibly dependent variables. This association is lowest for MATH, likely because the challenging nature of this task makes it difficult for the model to produce consistent answers.

*Table 6.* Somers' D computed between $\text{Acc}(y)$ and $\mathcal{V}(y)$ for $y \in \{y^+, y^-\}$ on test set.

| Dataset | Somers' D |
|---|---|
| GSM8K | 0.80 |
| MATH | 0.68 |
| ZebraLogic | 0.92 |

Furthermore, we measure the impact of the number of samples used to measure self-consistency ($k$) on its Somer's D correlation with correctness in Table 7. The results indicate that (i) lower values of $k$ (e.g. $k = 2/4$) have lower correlation with correctness or accuracy which we find is because of fewer instances where any answer gets multiple votes; (ii) while larger values of $k = 16$ yield slightly higher correlations, we prioritize computational efficiency in the data generation phase, and use a sufficiently large value of $k = 8$ in addition to filtering and a weighted loss in SCPO.

## B. Transduction During Inference

**Bootstrapping preference pairs from test queries further boosts performance.** In our primary experiments, we report results for two rounds of iterative training. However, as shown in Table 10, introducing a third round of SCPO yields only marginal improvements, with gains of less than 1% over the second round. To address this saturation, we explore generating new problems and building preference pairs using the queries from test split as exemplars instead of the train split. This strategy results in more substantial improvements (+1.44% for GSM8K), as it enables the model to better adapt to the unique characteristics of the test set. For MATH, we see more substantial improvements when using SC accuracy, resulting in an improvement bump of 1.26%. We note that ZebraLogic is excluded from this analysis, as it only provides test samples.

## C. Results on Math Reasoning with Llama-3.1

We now repeat the math reasoning experiments in Section 4.1 with Llama-3.1 Base 8B and find that while the absolute performance increases, the relative trends among the baselines remain the same – with SCPO$_{\text{Unsup.}}$ as the most performant unsupervised technique and SCPO$_{\text{Semi-Sup.}}$ yielding the overall highest accuracy on GSM8K and MATH. In Tables 8 and 9, we observe that two iterations

*Table 7.* Somers' D computed between $\text{Acc}(y)$ and $\mathcal{V}(y)$ for $y \in \{y^+, y^-\}$, i.e., the most and least consistent responses, on test set for different values of $k$.

| Dataset / Somer's D | $k = 2$ | $k = 4$ | $k = 8$ | $k = 16$ |
|---|---|---|---|---|
| GSM-8K | 0.39 | 0.65 | 0.80 | 0.89 |
| ZebraLogic | 0.66 | 0.82 | 0.92 | 0.93 |

*Table 8.* **GSM8K zero-shot accuracy** after training Llama-3.1 Base 8B with SCPO and baselines, using greedy or self-consistency (SC)-based inference.

| Method | Iter. | Train Data (K) | | Test Acc. (%) | |
|---|---|---|---|---|---|
| | | # Seed / Gen. | | Greedy | SC (8-way) |
| *without access to gold labels* | | / | | | |
| Seed model (zero-shot) | $M_0$ | - / - | | 43.14 | 59.59 |
| IRPO$_{\text{RM}}$ | $M_1$ | 6.5 / - | | 58.60 | 73.01 |
| | $M_2$ | 6.7 / - | | 60.04 | 72.19 |
| LMSI | $M_1$ | 6.7 / 5.7 | | 48.75 | 65.71 |
| | $M_2$ | 6.3 / 4.8 | | 52.39 | 60.42 |
| SCPO$_{\text{Unsup.}}$ | $M_1$ | 6.7 / 5.7 | | 61.64 | 71.95 |
| | $M_2$ | 5.5 / 4.9 | | **64.22** | **75.13** |
| *with access to gold labels* | | / | | | |
| IRPO$_{\text{Gold}}$ | $M_1$ | 5.6$^\dagger$ / - | | 60.05 | 76.04 |
| | $M_2$ | 5.8$^\dagger$ / - | | 65.50 | 79.61 |
| SCPO$_{\text{Semi-Sup.}}$ | $M_1$ | 5.6$^\dagger$ / 5.4 | | 65.60 | 79.08 |
| | $M_2$ | 5.2$^\dagger$ / 4.9 | | **68.46** | **79.75** |

*Table 9.* **MATH zero-shot accuracy** after training Llama-3.1 Base 8B with SCPO and baselines, using greedy or self-consistency (SC)-based inference.

| Method | Iter., | Train Data (K) | | Test Acc. (%) | |
|---|---|---|---|---|---|
| | | # Seed / Gen. | | Greedy | SC (8-way) |
| *without access to gold labels* | | / | | | |
| Seed model (zero-shot) | $M_0$ | - / - | | 15.70 | 24.62 |
| IRPO$_{\text{RM}}$ | $M_1$ | 6.2 / - | | 20.68 | 27.32 |
| | $M_2$ | 6.6 / - | | 20.74 | 25.88 |
| LMSI | $M_1$ | 0.9 / 0.9 | | 16.26 | 24.38 |
| | $M_2$ | 1.0 / 1.3 | | 15.94 | 22.60 |
| SCPO$_{\text{Unsup.}}$ | $M_1$ | 0.9 / 0.9 | | 19.38 | 27.74 |
| | $M_2$ | 1.4 / 1.7 | | **23.20** | **30.10** |
| *with access to gold labels* | | / | | | |
| IRPO$_{\text{Gold}}$ | $M_1$ | 2.7$^\dagger$ / - | | 22.40 | 31.64 |
| | $M_2$ | 3.2$^\dagger$ / - | | 22.86 | 32.30 |
| SCPO$_{\text{Semi-Sup.}}$ | $M_1$ | 2.7$^\dagger$ / 0.9 | | 22.98 | 32.18 |
| | $M_2$ | 3.2$^\dagger$ / 2.2 | | **24.36** | **32.64** |

of SCPO$_{\text{Semi-Sup.}}$ improve the greedy test accuracy of the seed model by 25.32% and 8.66% on GSM8K and MATH, respectively; while two iterations of SCPO$_{\text{Unsup.}}$ boost the greedy accuracy of the seed model by 21.08% on GSM8K and 7.5% on MATH dataset.

*Table 10.* Training $M_3$ by bootrapping from questions in the train and test set. On GSM8K, we bootstrap 8.7K, 5.8K pairs using train, and test problems, respectively. On MATH, we build 4.4K, and 4.2K preference pairs using train and test problems, respectively.

| Method | GSM8K Acc. | | MATH Acc. | |
|---|---|---|---|---|
| | Greedy | SC (8-way) | Greedy | SC (8-way) |
| $M_0$ | 41.17 | 58.80 | 14.46 | 18.20 |
| $M_1$ w/ SCPO$_{\text{Unsup.}}$ | 61.03 | **71.49** | 17.36 | 25.70 |
| $M_2$ w/ SCPO$_{\text{Unsup.}}$ | 63.91 | 71.11 | 19.72 | 24.58 |
| $M_3$ w/ SCPO$_{\text{Unsup.}}$ | 64.21 | 70.81 | 19.76 | 24.66 |
| $M_3$ w/ SCPO$_{\text{Unsup.}}$ on test queries | **65.35** | 70.96 | **20.00** | **25.84** |

# D. Prompts

We provide all task-specific prompts used for both generating new problems and for generating candidate solutions.

---

**Response Generation: ZebraLogic**

**Example Puzzle**:
There are 3 houses, numbered 1 to 3 from left to right, as seen from across the street. Each house is occupied by a different person. Each house has a unique attribute for each of the following characteristics:
- Each person has a unique name: 'Peter', 'Eric', 'Arnold'.
- Each person has a unique favorite drink: 'tea', 'water', 'milk'

## Clues:
1. Peter is in the second house.
2. Arnold is directly left of the one who only drinks water.
3. The one who only drinks water is directly left of the person who likes milk.

**Answer to the Example Puzzle**:
{
"reasoning": "Given Clue 1, we know Peter is in House 2. According to Clue 2, Arnold is directly left of the one who only drinks water. The person in House 3 cannot be on the left of anyone, so Arnold must be in House 1. Thus, Peter drinks water, and Eric lives in House 3. Then, according to Clue 3, Eric drinks milk. Therefore, Arnold drinks tea.",
"solution": { "House 1": { "Name": "Arnold", "Drink": "tea" },
"House 2": { "Name": "Peter", "Drink": "water" },
"House 3": { "Name": "Eric", "Drink": "milk" } }
}

**Puzzle to Solve**: {puzzle}
**Prompt:** Now please solve the above puzzle. Present your reasoning and solution in the following json format:
{json template}

---

**Response Generation: GSM8K**

**Prompt:** Answer the following question step-by-step. When you are ready, place the final answer in a new line as #### < number >.
**Q:** {question}
**A:** Let's think step by step.

---

---

**Response Generation: MATH**

**Prompt:** Answer the following question step-by-step. When you are ready, place the final answer in a new line as: The final answer is $\boxed{< your answer>}$
**Q:** {question}
**A:** Let's think step by step.

---

**Query Generation: ZebraLogic**

**Example Puzzle:**
*Attributes to Change*: ["Name", "Drink"]
"' There are 3 houses, numbered 1 to 3 from left to right, as seen from across the street. Each house is occupied by a different person. Each house has a unique attribute for each of the following characteristics:
- Each person has a unique name: 'Peter', 'Eric', 'Arnold'.
- Each person has a unique favorite drink: 'tea', 'water', 'milk'

## Clues:
1. Peter is in the second house.
2. Arnold is directly left of the one who only drinks water.
3. The one who only drinks water is directly left of the person who likes milk.
'"
**Answer:**
Let's change the "Name" and "Drink" attributes of the given puzzle to create a new puzzle. There are 3 names and drinks involved Mentions of "Name" changes from 'Peter', 'Eric', 'Arnold' to mentions of "Name": 'Molly', 'Shannon', 'Kelly' respectively. Instead of "Drink" as the attribute, let's their "Food" preferences as the attribute. So mentions of "Drink" changes from 'tea', 'water', 'milk' to mentions of "Food": 'pizza', 'burgers', 'fries'' respectively. Now, changing the language of the puzzle and clues we get,

*New Attribute Map*: {"Name": "Name", "Drink": "Food"}
*Puzzle*:
"' There are 3 houses, numbered 1 to 3 from left to right, as seen from across the street. Each house is occupied by a different person. Each house has a unique attribute for each of the following characteristics:
- Each person has a unique name: 'Molly', 'Shannon', 'Kelly'.
- Each person has a unique favorite food: 'pizza', 'burgers', 'fries'

## Clues:
1. Molly is in the second house.
2. Kelly is directly left of the one who only eats burgers.
3. The one who only eats burgers is directly left of the person who likes fries.
"'
**Puzzle to rephrase:**
*Attributes to Change*: {attributes dict}
"' {input puzzle} '"

**Prompt**: Rephrase the above puzzle by changing only the attributes above. ALWAYS mention the "New Attribute Map" and enclose the new puzzle within "' '". Aside from these attributes keep the logic of the puzzle as similar as possible. Similar to the example above, give your reasoning before rephrasing the puzzle.

---

Query Generation: GSM8K and MATH

**Q:** {few-shot question 1}
**Q:** {few-shot question 2}
**Q:** {few-shot question 3}
**Q:** {few-shot question 4}

**Prompt:** Based on the examples above, generate ONE solvable math word problem with similar difficulty. Note that all the information needed to solve the problem should be included in the question. Output the question and nothing else.
**Q:**

---

