# OpenReview forum: "Self-Consistency Preference Optimization"
_ICML.cc/2025/Conference — ICML 2025 poster_

### Official Review · Reviewer_rpC3 · 2025-03-11

**Overall Recommendation:** 3

**Summary:**

This paper introduces SCPO, a novel training method for large language models (LLMs) that leverages self-consistency to improve performance on complex reasoning tasks without requiring gold labels. SCPO iteratively trains models to prefer consistent answers over inconsistent ones by generating multiple responses, selecting the most and least consistent ones as preference pairs, and optimizing a weighted loss function based on the model's confidence in these pairs. The method is evaluated on GSM8K, MATH, and ZebraLogic datasets, showing significant improvements over existing unsupervised baselines and closing the gap with supervised training methods. Key findings include that SCPO outperforms models trained with external reward models and self-improvement methods, and it can further enhance results when combined with supervised learning. The approach demonstrates the effectiveness of using self-consistency as a training signal for improving both the accuracy and consistency of LLMs on reasoning tasks.

**Claims And Evidence:**

The claims made in the submission are generally supported by clear and convincing evidence. The authors provide extensive experimental results across multiple datasets (GSM8K, MATH, ZebraLogic) demonstrating performance improvements over baselines.

**Essential References Not Discussed:**

Some recent work on self-improving/self-annotating alignment also utilizes LLMs for preference annotation, similar to the approach. These references could support the rationale behind this work. For example:

1. Chen, Changyu, et al. "Bootstrapping language models with dpo implicit rewards." arXiv preprint arXiv:2406.09760 (2024).
2. Kim, Dongyoung, et al. "Spread Preference Annotation: Direct Preference Judgment for Efficient LLM Alignment." The Thirteenth International Conference on Learning Representations. 2025.

**Experimental Designs Or Analyses:**

The experimental designs are mostly sound but could benefit from including additional relevant baselines such as [1, 2]. These would provide more comprehensive comparisons and strengthen the validation of SCPO's effectiveness.


1. Chen, Changyu, et al. "Bootstrapping language models with dpo implicit rewards." arXiv preprint arXiv:2406.09760 (2024).
2. Kim, Dongyoung, et al. "Spread Preference Annotation: Direct Preference Judgment for Efficient LLM Alignment." The Thirteenth International Conference on Learning Representations. 2025.

**Methods And Evaluation Criteria:**

The proposed methods (SCPO) and evaluation criteria make sense for the problem of improving LLM reasoning without gold labels. SCPO's approach of using self-consistency to generate training signals aligns well with the challenge of lacking labeled data.

**Other Comments Or Suggestions:**

see `Strengths And Weaknesses`

**Other Strengths And Weaknesses:**

Strengths:
- The paper has intuitive motivation and relatively solid experimental validation.

Weaknesses:
- Limited baselines in experiments; should include more relevant comparison methods.
- No ablation studies on the number of responses k, which is crucial for the reweighting term w(x).
- The importance of the weighted SCPO loss is demonstrated in Table 5, but the discussion lacks references to specific related works like [1, 2, 3, 4, 5] where selecting high-quality data or weighting techniques are already common in DPO-based methods.

[1] $\beta$-dpo: Direct preference optimization with dynamic $\beta$. NeurIPS, 2024.
[2] Preference optimization with the pairwise cringe loss. arXiv preprint arXiv:2312.16682.
[3] Reward difference optimization for sample reweighting in offline rlhf. EMNLP 2024.
[4] Rs-dpo: A hybrid rejection sampling and direct preference optimization method for alignment of large language models. EMNLP 2024.
[5] Filtered direct preference optimization. EMNLP 2024.

**Questions For Authors:**

see `Strengths And Weaknesses`

**Relation To Broader Scientific Literature:**

N/A

**Theoretical Claims:**

N/A

---

> ### Author Rebuttal · Authors · 2025-04-01
>
> We thank you for your detailed review and comments. We are also glad to see you found our paper to have “intuitive motivation” as well as “extensive experimental results”. Please find our in-depth response to your comments below.
>
> > Additional relevant baselines:
>
>  Thanks for pointing these papers out to us, we contrast ScPO to these works as:
>
> - **Chen et al. 2024**: We argue that they are focused on **general instruction following, and their method is not proven to work in reasoning tasks** (as we do in ScPO) like math etc. Also, note that their main technical contribution is to add a length control factor to DPO’s implicit reward term. While “length bias” has been an issue with general instruction following tasks [(Dubois et al 2024](https://arxiv.org/abs/2404.04475)) as models spuriously prefer longer generations, preference evaluation for reasoning domains is more straightforward – preferring correct answers and rejecting the rest, so we expect length-controlled DPO to perform similarly to the standard DPO algorithm used in ScPO.
> - **Kim et al. 2025**: At its core, this paper uses the **model's confidence on two generations corresponding to the same prompt, to construct preference pairs** with additional preference label smoothening for the bottom 10% annotations that are expected to be noisy. This stands in contrast to ScPO which uses self-consistency of predictions to generate preference annotations, and we compare the quality of the two metrics (confidence vs self-consistency) to curate preference data below to demonstrate the superiority of ScPO on reasoning tasks:
>   - *Setup:* We incorporate Kim et al.’s strategy for preference data creation by using the model’s confidence. Specifically, we compute the confidence for each generation (from the same sample set used to compute SC) and create preference pairs by choosing the most confident response and rejecting the least confident response. Further, following Kim et al. (2025), we filter out the last 10% of training instances with the lowest difference in confidence of chosen and rejected responses (indicative of noise in annotation), and compare its Somer’s D correlation with accuracy with that of self-consistency (Table 6).
>
> | Dataset/Metric | Confidence (Kim et al. 2025\) | Self-Consistency (ours) |
> | :---- | :---- | :---- |
> | GSM8K | 0.11 | 0.80 |
> | ZebraLogic | \- | 0.93 |
>
>   - *Results:* On ZebraLogic, we find that the accuracy of the most confident answer is in fact 0 rendering Kim et al.’s method completely ineffective, whereas SC on the other hand correlates strongly with correctness. Similarly, on GSM8K we find that confidence score of generation shows little correlation with correctness, and is most likely not effective to generate preference training data. **The results indicate that across datasets self-consistency offers a better estimate of correctness of an answer, and therefore yields higher quality preference for training.** These findings are consistent with prior work that uses self-consistency to estimate model confidence for reasoning tasks ([Xiong et al. 2023](https://arxiv.org/abs/2306.13063), [Kabra et al. 2023](https://arxiv.org/pdf/2311.09553)).
>
> > Ablation with number of samples (K)
>
> **We measure the impact of the number of samples used to measure self-consistency on its Somer’s D correlation with correctness (as done in Table 6\) for K=2,4,8,16.**
>
> | Dataset | K=2 | K=4 | K=8 | K=16 |
> | :---- | :---- | :---- | :---- | :---- |
> | GSM-8K | 0.39 | 0.65 | 0.80 | 0.89 |
> | ZebraLogic | 0.66 | 0.82 | 0.92 | 0.93 |
>
> The results indicate that (i) lower values of K (e.g. K=2/4) have lower correlation with correctness or accuracy which we find is because of fewer instances where any answer gets multiple votes; (ii) while larger values of K=16 yield slightly higher correlations, we prioritize computational efficiency in the data generation phase (L100-122), and use a sufficiently large value of K=8 in addition to filtering (L118-121) and a weighted loss (L141-149) for ScPO training.
>
> > related works like \[1, 2, 3, 4, 5\] where selecting high-quality data or weighting techniques are already common in DPO-based methods
>
> Thank you for pointing out these works. We haven’t discussed them in our paper because they are about weighting prompts based on their quality, removing noisy or unhelpful questions. In contrast, our work is about weighing solutions, giving less importance to solutions that are likely to be wrong. However, it could be helpful to discuss those works and their differences, which we will do in the final revision.
>
> We hope these additional results and explanations address your questions and will allow you to revisit your score.

---

> > ### Comment · Reviewer_rpC3 · 2025-04-03
> >
> > Thank you for your efforts and response. I will increase my score accordingly.
> >
> > Additionally, I believe the final version should include a discussion and comparison of weighting methods in preference learning, as this would further strengthen the manuscript.

---

> > > ### Author Response · Authors · 2025-04-03
> > >
> > > Thank you for your continued engagement and for revisiting your score. We will add a discussion and comparison to the papers you pointed out in the final paper.

---

### Official Review · Reviewer_Z9n6 · 2025-03-13

**Overall Recommendation:** 3

**Summary:**

This paper considers self-alignment of LLMs, where the data consist of only prompts but not the ground truth. The authors propose to use the self-consistency to choose winning and losing samples, where the responses corresponding to the most/least final answers are considered as the winning/losing samples.

**Claims And Evidence:**

The major claim is that self-consistency preference optimization improves self-alignment for reasoning task. This claim is quite intuitive and plausible. Their results served as empirical evidence for this claim.

**Essential References Not Discussed:**

Another concern is that this work is very similar to [2]. See their Figure 2 for their method with self-consistency feedback. Since [2] is made public last Nov, I consider this as a concurrent work to this paper. My positive evaluation is based on the consideration that both works are concurrent (so it did not hurt the score of my evaluation).

[2] Preference Optimization for Reasoning with Pseudo Feedback. https://arxiv.org/abs/2411.16345v1

**Experimental Designs Or Analyses:**

The baselines are reward-model fine-tuning, and language model self-improving (LMSI) [1]. The choices are to certain extent reasonable, one concern here is that the self-improvement baseline, LSMI, is not quite up-to-date. Considering self-alignment as a quite active area, there should be more follow-ups suitable for serving as a baseline.

[1] Large Language Models Can Self-Improve. https://arxiv.org/abs/2210.11610

**Methods And Evaluation Criteria:**

The method itself is quite intuitive and simple, which is likely to be effective for unsupervised self-alignment, as self-consistency is well-recongnized for filtering samples.

The method and baselines are evaluated on popular reasoning datasets, GSM8K and MATH, which is quite standard hence I have no question about the evaluation.

**Other Comments Or Suggestions:**

see above

**Other Strengths And Weaknesses:**

see above

**Questions For Authors:**

see above

**Relation To Broader Scientific Literature:**

N/A

**Theoretical Claims:**

N/A

---

> ### Author Rebuttal · Authors · 2025-04-01
>
> We thank you for your review and for appreciating the “intuitive and simple” design of our method as well as our evaluation setup.
>
> > The choices are to certain extent reasonable, one concern here is that the self-improvement baseline, LSMI, is not quite up-to-date. Considering self-alignment as a quite active area, there should be more follow-ups suitable for serving as a baseline.
>
> We are not aware of a more up-to-date follow-up of the LMSI baseline that is applicable to our unsupervised setting but please let us know if we missed something you had in mind. Furthermore, we note that two variants of IRPO as well as the 8B RM are fairly up to date since **IRPO was published in Neurips 2024, and the Armo RM (also released in mid 2024\)** was among the best performing 8B reward model as per the RewardBench leaderboard at the time of development.
>
> > Another concern is that this work is very similar to \[2\]. See their Figure 2 for their method with self-consistency feedback. Since \[2\] is made public last Nov, I consider this as a concurrent work to this paper.
>
> While the “pseudo feedback from self-consistency” idea in \[2\] is concurrent and related to ScPO, we would like to identify the following key differences:
>
> - As shown in Fig 1 (left) and Sec 2 (L 81-99), we show that ScPO can be used to augment the initial seed training data with additional questions sampled from the same model. Furthermore, we show that self-consistency plays a crucial role in filtering for well-formed and answerable questions, which \[2\] lacks. The effectiveness of generating additional data can be seen from not only the unsupervised results, but also our results from the semi-supervised setting where ScPO outperforms the supervised IRPO (gold) baseline by up to 2.35% on GSM8K.
> - Next, we also incorporate self-consistency in our weighted loss that adjusts the weight of a training instance based on the relative confidence, i.e. difference in vote share of chosen and rejected responses (L141-149). We demonstrate the importance of this weighted training loss in Table 4 of Sec 5\. Thus, we argue that \[2\] is equivalent to the w(x)=1 baseline (only trained on seed data) which ScPO outperforms.
>
>  That being said, we thank you for pointing out this concurrent work and we will cite it in future versions of our paper.

---

### Official Review · Reviewer_R9NR · 2025-03-13

**Overall Recommendation:** 4

**Summary:**

The paper introduces Self-Consistency Preference Optimization (SCPO), a novel approach to self-training large language models (LLMs) for complex reasoning tasks without requiring gold labels/solutions. SCPO extends the concept of self-consistency (typically used only at inference time) to create preference pairs during training by sampling multiple responses for each problem and identifying the most consistent vs. least consistent answers. The key innovation is a weighted preference optimization objective where weights depend on the vote margin between chosen and rejected responses. The paper also presents a semi-supervised variant that incorporates gold labels when available, further improving performance.

The authors validate the method on reasoning (GSM8K, MATH) and logical reasoning (ZebraLogic) benchmarks, showing that unsupervised SCPO nearly matches supervised preference optimization with gold labels, while semi-supervised SCPO outperforms fully supervised baselines. Most notably, on ZebraLogic, SCPO helps Llama-3 8B outperform significantly larger models like Llama-3 70B and Claude-3 Haiku.

**Claims And Evidence:**

The claims in this paper are supported by empirical evidence. The primary claim that SCPO improves reasoning abilities without access to gold solutions is demonstrated through significant performance gains on GSM8K, MATH, and ZebraLogic. The claim that unsupervised SCPO approaches the performance of supervised training is supported by results showing <1% gap in performance. The authors also provide evidence for the superiority of their weighted loss function through ablation studies that show 1-2.5% improvements over unweighted alternatives.

The paper includes analyses showing the correlation between vote share and accuracy, which validates the main assumption that consistency is a good proxy for correctness although this result was relegated to Appendix A.

**Essential References Not Discussed:**

The paper covers the most relevant literature well. I don't see any critical omissions that would significantly impact the understanding of the work's context or contributions. The authors cite key papers on self-consistency, preference optimization, and self-training approaches.

**Experimental Designs Or Analyses:**

I examined the core experiments in the paper and found no issues:
1. Baseline comparisons on reasoning datasets (Tables 1-3): sound evaluation of the core points of the paper.
2. Weighted loss ablation (Table 4): Direct comparison between weighted and unweighted versions.
3. Consistency analysis (Figure 2): Measurement of vote share increases across iterations.
4. Threshold filtering experiment (Table 5): Testing different consistency thresholds.
5. Preference accuracy analysis (Figure 3): Comparison of self-consistency vs. reward models in correctly ordering preferences.

The hyperparameters ar well-documented (learning rate, epochs, temperature settings). The only limitation is the lack of ablation on the number of samples (k) used for voting, which would help understand computational efficiency tradeoffs. Otherwise, the experiments provide strong evidence for the paper's claims.

**Methods And Evaluation Criteria:**

The use of GSM8K and MATH as benchmarks for mathematical reasoning and ZebraLogic for logical reasoning is a sensible choice. The authors evaluate using both greedy decoding and self-consistency inference, showing improvements in both settings. The comparison against multiple baselines (zero-shot CoT, IRPO_RM, LMSI, IRPOGold) is broad and provides enough evidence for the effectivenetss of SCPO.

It would be good to use benchmarks thay go beyond math and logical reasoning to show the versitility of SCPO but those are notoriously difficult to come by. The focus on GSM8K, MATH, and ZebraLogic is understandable.

**Other Comments Or Suggestions:**

- It would be valuable to analyze how SCPO affects the diversity of reasoning paths across iterations
- A more detailed analysis of computational requirements compared to baselines would help readers understand practical tradeoffs

**Other Strengths And Weaknesses:**

Strengths:
- Clear presentation of the work makes understanding the paper very easy
- Creative application of self-consistency to training rather than just inference
- Impressive results on ZebraLogic, showing a smaller model can outperform much larger ones
- Strong empirical validation across multiple datasets and model scales
- Thoughtful ablation studies that justify design choices

Weaknesses:
- Limited discussion of computational overhead from generating multiple responses per query during training
- Current evaluation limited to math and logic tasks with definitive answers; unclear generalizability to more open-ended reasoning tasks
- While the paper shows that consistency correlates with correctness, deeper analysis of when/why this relationship might break down would strengthen the work

**Questions For Authors:**

1. Have you investigated the trade-off between computation costs and performance gains in SCPO? Specifically, how does the computational overhead of generating multiple responses during training compare to other methods, and how might this scale with larger models?

2. The results show diminishing returns after the second iteration of SCPO. Do you have insights into whether this is due to fundamental limitations of self-consistency as a training signal, or might there be ways to continue improving with more iterations?

3. For real-world applications, have you considered how SCPO might perform on problems where there isn't a single definitive correct answer, or where answer extraction is more challenging than in the studied tasks?

**Relation To Broader Scientific Literature:**

SCPO extends Wang et al.'s (2023) self-consistency concept from inference to training, presenting a clean and practical strategy for self-improvement in LLMs. While methods like LMSI (Huang et al., 2023) also use self-consistency for unsupervised training, SCPO builds on them improving the end performance. The approach relates to self-alignment work like Yuan et al. (2024), but addresses their limitation in reasoning tasks identified by Huang et al. (2024).

**Theoretical Claims:**

The paper doesn't present formal theoretical claims or proofs. It focuses on the presentation and empirical validation of their new method.

---

> ### Author Rebuttal · Authors · 2025-04-01
>
> We thank you for your extensive review and comments and are glad to see you appreciate the “creative application of self-consistency”, “impressive results”, and “thoughtful ablations”. Please find our detailed response to your comments below and let us know if you have any follow up questions.
>
> > Impact of number of samples (K)
>
> **We measure the impact of the number of samples used to measure self-consistency on its Somer’s D correlation with correctness (as done in Table 6\) for K=2,4,8,16.**
>
> | Dataset | K=2 | K=4 | K=8 | K=16 |
> | :---- | :---- | :---- | :---- | :---- |
> | GSM-8K | 0.39 | 0.65 | 0.80 | 0.89 |
> | ZebraLogic | 0.66 | 0.82 | 0.92 | 0.93 |
>
> The results indicate that (i) lower values of K (e.g. K=2/4) have lower correlation with correctness or accuracy which we find is because of fewer instances where any answer gets multiple votes; (ii) while larger values of K=16 yield slightly higher correlations, we prioritize computational efficiency in the data generation phase (L100-122), and use a sufficiently large value of K=8 in addition to filtering (L118-121) and a weighted loss (L141-149) for ScPO training.
>
> > Computational Overhead of ScPO:
>
> We reiterate that **all our baselines including IRPO, LMSI, as well as ScPO have the same computational overhead by design** as we use similar size training datasets, same number of samples (K), and the same training hyperparameters. Specifically, to your point on generating training data, we note that this process is done once at the start of each iteration when using the LLM at inference-time, and *not during training*. Therefore, we can make use of popular strategies for speeding up LLM inference such as the [vLLM library](https://github.com/vllm-project/vllm), increasing batch-size, and utilizing multiple GPUs in parallel, making the data-generation process far less computationally demanding than the training itself. We will include this discussion in the paper.
>
> > Impact of ScPO on model diversity:
>
> **In Fig 2 of Sec 5, we visualize the vote share of the model's responses across iterations, and find that it increases across iterations for all datasets**, indicating a decrease in the number of unique answers (diversity). We suspect this is a consequence of RLHF training that has been well-documented in prior work ([Kirk et al. 202](https://arxiv.org/abs/2310.06452)4, [Murthy et al. 2024](https://arxiv.org/abs/2411.04427), Murthy et al. 2024\) and is outside the scope of our study. At the same time, in Sec 4 (Tables 1-3 and 8-9), we report test accuracy after 8-way self-consistency and find that models trained with ScPO continue to benefit from diversity in generations via SC at test-time.
>
> > Number of Iterations:
>
>  We refer you to Table 7 in Appendix B where we find that performance on math reasoning largely plateaus after the second iteration, with \<1 point gain with a third iteration on training. However, in the same table we find that sampling questions from a different distribution, i.e., the distributions of questions (*without using the answers*) from the test set and using it to sample additional related problems (L81-98) yields additional improvements in the third iteration on top of the M\_2 models (L270-274). Therefore, we believe that increasing the diversity of the seed problems, or after each iteration can be an effective way to delay performance saturation. Also, developing a RLHF method that does not diminish diversity can be a solution.
>
> > Open Ended Reasoning Tasks:
>
> We reiterate that ScPO is designed to improve model’s reasoning performance with unsupervised or semi-supervised training. Different from general instruction following tasks such as creative writing with subtle human preferences, on reasoning tasks we desire to prefer correct solutions and disprefer incorrect ones. Nevertheless, in such scenarios, we believe ScPO can be combined with techniques to measure consistency in more generative settings such as universal self-consistency (L432-436) or using executable programs to measure correctness ([Lambert et al. 2024](https://arxiv.org/abs/2411.15124)). Note that we use a similar programmatic or symbolic approach to measure consistency **for the ZebraLogic benchmark where the answer is in a complex “json” format that cannot be directly compared via exact string match and requires a symbolic function to measure equivalent answers and multiple votes**. Measuring consistency in more open-ended tasks (e.g. creative writing) is indeed under explored, and likely to be first worked out in inference time uses, which we leave for future work. We will expand on the final paper.

---

> > ### Comment · Reviewer_R9NR · 2025-04-04
> >
> > Dear Authors,
> >
> > Thank you for your responses. I think that those answers reinforce my score and I would like to see the paper accepted.
> >
> > Best

---

### Official Review · Reviewer_XcBF · 2025-03-14

**Overall Recommendation:** 3

**Summary:**

The paper introduces Self-Consistency Preference Optimization (ScPO), an unsupervised method for training LLMs to improve reasoning tasks. ScPO leverages the concept of self-consistency—traditionally used at inference—to iteratively optimize models by preferring answers with high consensus over inconsistent ones.

- A weighted loss function that prioritizes high-confidence preference pairs based on vote margins.

- Semi-supervised extensions combining labeled and unlabeled data.

- Experiments on GSM8K, MATH, and ZebraLogic showing ScPO outperforms supervised baselines (e.g., +22.74% on GSM8K) and larger models (e.g., Llama-3 8B trained with ScPO surpasses Llama-3 70B on ZebraLogic).

**Claims And Evidence:**

Most claims are supported by empirical results. ScPO outperforms IRPO and LMSI baselines (Tables 1–3) but the improvement is not significant. But the evaluation just considers zero-shot accuracy, and I'm not sure whether other baseline models are using the best configurations, or if other models will perform better using inference-time weighted voting/reward model.

**Essential References Not Discussed:**

N/A

**Experimental Designs Or Analyses:**

Strengths: Ablation studies (Tables 4–5) and correlation analysis (Appendix A) strengthen validity.

Weaknesses:
- The choice of 2 iterations is under-explained (mentions some related work but lacks systematic analysis).
- Threshold τ is tuned on dev sets, but sensitivity to this hyperparameter is not thoroughly tested.

**Methods And Evaluation Criteria:**

Make sense. The benchmark datasets GSM8K and Math are adequate.

**Other Comments Or Suggestions:**

N/A

**Other Strengths And Weaknesses:**

see above

**Questions For Authors:**

1. How does ScPO generalize to tasks with ambiguous final answers?
2. Could higher vote shares reflect over-confidence rather than accuracy? How is this risk mitigated?

**Relation To Broader Scientific Literature:**

ScPO builds on self-consistency and preference optimization, and may contribute to improve LLM's self-consistency for math reasoning.

**Theoretical Claims:**

No theoretical proofs are provided. The loss function is empirically justified but lacks formal analysis (e.g., convergence guarantees or why vote margins correlate with correctness).

---

> ### Author Rebuttal · Authors · 2025-04-01
>
> We thank you for your detailed review and questions. Please find our response below your comments:
>
> > Significance of Results
>
> Our test sets include \>= 1K samples for GSM8K and ZebraLogic, and 5K problems for MATH. In our primary unsupervised paradigm with greedy decoding, ScPO **consistently** outperforms the IRPO (RM) and LMSI baselines across **three datasets and two base models** *(Tables 1-3, 8-9)*. The unsupervised baselines exhibit high variance across different datasets and models. For example, while LMSI is the second-best unsupervised method behind ScPO (by 7.2%) with Llama-3 8B in Table 1, it performs significantly worse with Llama-3.1 8B on the same GSM8K dataset in Table 8, trailing ScPO by 11.83% and IRPO RM by 7.6%. Additionally, IRPO RM is the least effective on GSM8K and ZebraLogic with Llama-3 8B. Given that all methods have similar computational and data budgets (see Sec 4), this strongly supports the effectiveness of ScPO for reasoning tasks.
>
> > Configuration of Baselines
>
> We conduct zero-shot evaluation to assess the improvement in the model's reasoning abilities from zero-shot instruction training. As shown in Tables 1-3, 8-9, using 8-way self-consistency (SC) on the test set, SC improves the performance of all baselines. However, models trained with ScPO still achieve the highest performance (after SC) in both supervised and semi-supervised settings across two datasets and two model families. Therefore, we expect similar results for other inference-time variants, such as weighted SC or Best-of-N sampling.
>
> > Vote-margin and Correctness
>
> We reiterate that the intuition behind self-consistency is that model errors are generally random, making it unlikely to repeat the same incorrect answer across different samples (L 28-36). This concept, effective in various domains and predating its use in LLMs (e.g., RANSAC by Fischler & Bolles, 1981), is supported by popular LLMs showing improved accuracy with majority voting in reasoning tasks (e.g., e.g. DeepSeekMath, Gemini1.5). Empirically, our results (Appendix A, Table 6\) demonstrate a strong correlation between consistency and correctness across datasets, indicating LLMs are not widely over-confident in math and logical reasoning tasks. This aligns with prior findings that LLMs are well-calibrated ([Kadavath et al. 2022](https://arxiv.org/abs/2207.05221)) and that self-consistency reliably estimates model confidence ([Xiong et al. 2023](https://arxiv.org/abs/2306.13063), [Kabra et al. 2023](https://arxiv.org/pdf/2311.09553)). In domains with prevalent overconfidence, self-consistency could be combined with inference-time calibration techniques ([Zhao et al. 2021](https://arxiv.org/abs/2102.09690)).
>
> > Number of Iterations
>
> Refer to Table 7 in Appendix B, where we observe that math reasoning performance largely plateaus after the second iteration, with less than a 1-point gain in the third iteration. However, the same table shows that sampling questions from a different distribution—using the distributions of questions (*without answers*) from the test set to sample additional related problems (L81-98)—yields further improvements in the third iteration on top of the M2 models (L270-274).
>
> > Sensitivity to Threshold τ
>
> As noted in L352-378 and Table 5, the initial threshold is based on the training data quality (Margin: Acc(preferred) \- Acc(rejected)) and the number of instances meeting the cutoff, i.e., Vote(preferred) \>= τ. Even at lower thresholds, Table 5 shows we can improve the base model's performance. While the training data quality and sample size depend on the LLM's inherent consistency for a specific domain/dataset, we applied the same method to train the Llama-3.1 8B model in Appendix C and achieved significant gains without tuning hyperparameters for the new base model.
>
> > Dealing with Ambiguous Answers
>
> We reiterate that ScPO is designed to improve model’s reasoning performance with unsupervised or semi-supervised training. Different from general instruction following tasks such as creative writing with subtle human preferences, **on reasoning tasks we desire to prefer correct solutions and disprefer incorrect ones, making the domain relatively less ambiguous** (please let us know if you have any specific dataset in mind). Nevertheless, ScPO can be combined with techniques to measure consistency in generative settings, such as universal self-consistency (L432-436) or executable programs for correctness ([Lambert et al. 2024](https://arxiv.org/abs/2411.15124)). For example, in the ZebraLogic benchmark, we use a programmatic approach to measure consistency, where answers in complex "json" format require a symbolic function for comparison and multiple votes.
>
> We hope our response has addressed all of your questions and will allow you to revisit your score. We are happy to answer any followup questions and requests you may have.

---

### Decision · Program_Chairs · 2025-05-01

**Decision:**

Accept (poster)

**Comment:**

This paper proposes an unsupervised method named ScPO, which can improve LLMs' ability of complex reasoning without requiring gold labels.

Most reviewers acknowledged the motivation, the intuitiveness of the method, the rationality of the experiments, the validity of the results and writing quality of the submission. The authors make great efforts and address most of the concerns of the reviewers during the rebuttal phase. All the reviews tend to accept this paper during the reviewer-author discussion phase.

Overall, this paper is valuable for the community. Therefore, I recommend accepting this paper.